# A New Sinamiin Fish (Actinopterygii) from the Early Cretaceous of Thailand: Implications on the Evolutionary History of the Amiid Lineage

Uthumporn Deesri [1,*], Wilailuck Naksri [2], Pratueng Jintasakul [2], Yoshikazu Noda [3], Hirokazu Yukawa [3], Tamara El Hossny [4,5] and Lionel Cavin [5]

1  Department of Biology, Faculty of Science, Mahasarakham University, Maha Sarakham 44150, Thailand
2  Northeastern Research Institute of Petrified Wood and Mineral Resources, Nakhon Ratchasima Rajabhat University, Nakhon Ratchasima 30000, Thailand
3  Fukui Prefectural Dinosaur Museum, 51-11 Muroko, Terao, Katsuyama 911-8601, Fukui, Japan
4  Department of Earth Sciences, University of Geneva, Rue des Maraîchers 13, 1205 Geneva, Switzerland
5  Department of Geology and Palaeontology, Natural History Museum of Geneva, CP6434, 1211 Geneva, Switzerland
*  Correspondence: uthumporn_deesri@yahoo.com

**Abstract:** The Sinamiidae are a family of halecomorph fishes (Holostei) stratigraphically limited to the Lower Cretaceous and confined to East Asia. The first species of sinamiids were discovered in China, and then new occurrences were recorded in Thailand and Japan. The three recognized genera, *Sinamia*, *Siamamia* and *Ikechaoamia*, are notably characterized by an unpaired parietal. Here, we describe a new genus and species of sinamiid based on material from the Aptian Khok Kruat Formation of Ban Krok Duean Ha, Nakhon Ratchasima, Thailand. The new taxon known from preserved specimens in 3D is characterized by four pairs of extrascapular and tall cylindrical teeth with a conical enamel stalk topped by an arrowhead-shaped acrodine cap, among other characters. A phylogenetic analysis of the halecomorph fishes shows that the new taxon is the sister of the other Thai species, *Siamamia naga*, and that the two are grouped with two Chinese genera in a strongly supported clade, the Sinamiinae. This subfamily is here grouped with the Amiinae that contained the extant *Amia*. This new discovery is a clue that Southeast Asia may have been a center of diversification for this fish clade, and the phylogenetic analysis reveals that amiines may have originated somewhere in Asia during the Cretaceous before they spread throughout the northern hemisphere.

**Keywords:** Halecomorphi; Amiidae; Sinamiinae; phylogeny; paleobiogeography; Khorat Plateau; Early Cretaceous; *Khoratamia phattharajani*

## 1. Introduction

### 1.1. The Sinamiinae

Amiiform fishes are the only extant representatives of the Halecomorphi, a clade of neopterygians grouped with the Ginglymodi among the Holostei, and represented today by two species in freshwaters of North America, *Amia calva* and *A. aucellicauda* [1,2]. The Amiidae contains the extant Amiinae, known in the fossil record from the terminal Cretaceous, and several Late Jurassic and Cretaceous extinct subfamilies, i.e., Vidalamiinae, Solnhofenamiinae and Amiopsinae. In their monography of the Amiidae, Grande and Bemis [3] found that the sister clade of the Amiidae is the Sinamiidae. These phylogenetic relationships have been found in most subsequent analyses since a study by Xu [4] who, based on a new synthetic datamatrix, found sister relationships between the Amiinae (*Amia* and *Cyclurus*) and the Sinamiinae. These new relationships are important because it has strong palaeobiogeographical implication for these two freshwater lineages of amiiforms.

The first described sinamiid is †*Sinamia zdanskyi* Stensiö, 1935 [5], which was discovered in the Early Cretaceous Mengyin Formation in Shandong province, China. This species was then reported in Gansu, Neimenggu, Ningxia, Shaanxi and Shandong provinces [6]. Other species of *Sinamia* have been subsequently described from the Early Cretaceous of China, i.e., †*S. huananensis* Su, 1973 [7], in Anhui and Zhejiang; †*S. chinhuaensis* Wei, 1976 [8], in Zhejiang; †*S. poyangica* Su and Li, 1990 [9], in Jiangxi; †*S. liaoningensis* Zhang, 2012 [10], in Liaoning; and †*S. lanzhouensis* Peng et al., 2015 [11] in Gansu. Another genus of sinamiid, †*Ikechaoamia*, was originally reported in Neimenggu with †*I. orientalis* Liu, 1961 [12], followed by another species from Zhejiang, †*I. meridionalis* Zhang and Zhang, 1980 [13]. Outside China, †sinamiid remains have been described from the Wakino subgroup in Japan and Nagdong subgroup in Korea by Yabumoto and colleagues [14], and *S. kukurihime* Yabumoto, 2014 [15] described from the Berriasian–Hauterivian Kuwajima Formation of the Tetori Group in Japan. In Thailand, the first sinamiid fish was referred to a new genus, *Siamamia*, with *S. naga* Cavin et al., 2007 [16], described based on partly articulated skulls and many isolated ossifications from the Phu Phok locality localized in the Berriasian–Barremian Sao Khua in the northeastern part of the country. Since then, isolated remains of sinamiid have been recorded in several localities of the Sao Khua Formation and from the younger Aptian–Albian Khok Kruat Formation [17,18].

Here, we report new, well-preserved sinamiid materials found in the locality of Ban Krok Duean Ha, Nakhon Ratchasima, localized in the Khok Kruat Formation. The objective of the study is to describe the new taxon, which belongs to a new genus and species, and to compare it with other species of sinamiins and other amiiform taxa. A phylogenetic analysis is conducted, but it does not take into account all species of *Sinamia*, the relationships of which have been studied by Yabumoto [19].

*1.2. Geological and Paleoenvironmental Settings*

The Khorat Group outcrops across the Khorat Plateau of northeast Thailand and extends east to Laos and south to Kampuchea. The Khorat Group comprises five formations, with the Khok Kruat Formation at the top of the series. The Khok Kruat Formation consists of reddish-brown fine- to medium-grained sandstone, siltstone and mudstone, with some conglomerate beds [20]. This formation is predominantly fluvial in origin [21].

The study area is located in Ban Krok Duean Ha, Suranaree subdistrict of Nakhon Ratchasima Province, northeast Thailand (Figure 1). Since 2017, the Northeastern Research Institute of Petrified Wood and Mineral Resources Rajabhat University and the Fukui Prefectural Dinosaur Museum have been excavating at this dinosaur site (KDH). The fossiliferous strata were originally covered with soil about several tens of centimeters thick. After removal of the soil, the bedrock appears as a mosaic of large blocks of sandstone forming kinds of "islands". The vertebrate assemblage mainly includes isolated bones of dinosaurs, pterosaurs, fish, crocodiles and turtles, together with coprolites.

The sedimentary succession of the excavation site is composed of medium- to coarse-grained sandstone and conglomerate. Mudstones are rare. The conglomerate consists mainly of clay rip-up pebble clasts and rounded calcareous nodule granules. Lithological and sedimentological features (planar and trough cross stratified medium- to coarse-grained sandstone with nodule granule and clay rip-up pebble bed, some beds eroded lower sediments) indicate that sediments were deposited in channels and bars. The fish specimens were found in vaguely parallel strata containing relatively well-sorted coarse-grained sandstone, clay rip-up pebble and nodule granule beds (Figure 2).

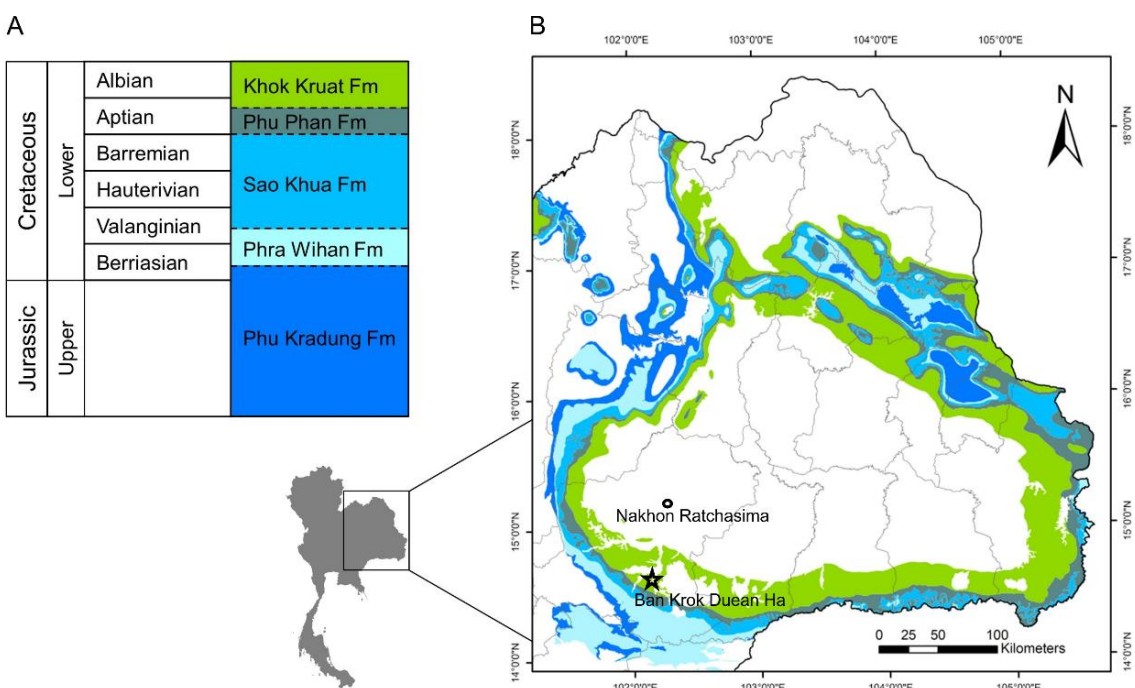

**Figure 1.** (**A**) Stratigraphy of the Khorat Group. (**B**) Map showing the formations of the Khorat Group, Khorat Plateau, northeastern Thailand, with the location of Ban Krok Duean Ha indicated (star). (Modified from Manitkoon et al. [22]).

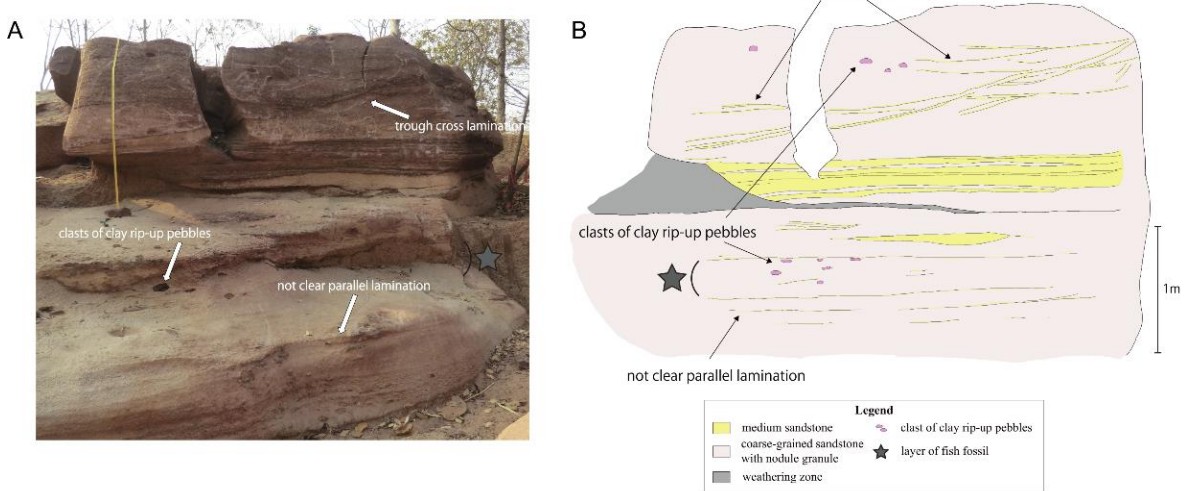

**Figure 2.** (**A**) The section of the sample collection and (**B**) stratigraphic section of the study area.

## 2. Materials and Methods

### 2.1. Examined Material

The material described here was found at the locality of Ban Krok Duean Ha, Nakhon Ratchasima Province (Figure 1), during a Thai–Japan joint excavation in 2018. The holotype (NRRU-F01020023), paratype (NRRU-F01020024) and a patch of scales (NRRU-F01020025) are housed at the Northeastern Research Institute of Petrified Wood and Mineral Resources. The holotype, KT-181, is a 3D-preserved anterior half of an individual. It was broken during the field collection and glued in the laboratory. The paratype (KR-11) is preserved as a partly articulated postcranial skeleton including the squamation and part of the vertebral column, but with the fin missing because of preservation. The paratype is a bigger specimen than the holotype by comparison with dimension of its scales. During the field excavation, the specimens were extracted using hammer to break down the rock. As a consequence,

the material consists of isolated pieces, which were glued afterward. The preparation was performed using an air pen in the laboratory of the Paleontological Research and Education Center, Mahasarakham University. An air-pen with a pointed chisel 4 mm sized was used for operating the sediment far from the fossil, and then a needlepoint and a small air scribe with lower pressure were used to free the surface of the fossil. The time-consuming preparation was performed under a binocular scope.

The nomenclature used in the anatomical description follows Grande and Bemis [3].

### 2.2. Phylogenetic Analysis

To establish the phylogenetic position of this new amiiform within the Halecomorphi, we chose the data matrix of Xu [4] as it is one of the most recent compilations of neopterygian characters. However, since the focus is on the position of *Khoratamia* (and *Siamamia*) within the Halecomorphi, we removed the 32 teleost and ginglymodian taxa. We used the same outgroups as in Xu [4], *Pteronisculus stensioi* and *Teffichthys madagascariensis*, and we added data for *Khoratamia phattharajani* and *Siamamia naga* from the present study, in addition to another ionoscopiform, *Spathiurus dorsalis*, from El Hossny et al. [23]. We first ran a preliminary analysis using the 224 characters of Xu [4] for these 29 in-group taxa. The result of a heuristic search showed that 105 characters were uninformative and hence removed for the main analysis. All the character states and the 119 parsimony informative characters retained were treated as unordered and equally weighted, and multistate characters were treated as polymorphisms.

### 2.3. Anatomical Terminologies

Anatomical terminologies follow: ang, angular; ao, antorbital; br, branchiostegal rays; c, centrum; cha, anterior ceratohyal; chp, posterior ceratohyal: cl, cleithrum; d, dentary; dpt, dermopterotic; dsp, dermosphenotic; ecptp, ectopterygoid tooth patch; enp, entopterygoid; es, extrascapular; exo, exoccipital; fm, foramen magnum; fr, frontal; g, gular; h, hyomandibula; hh, hypohyal; hmf, foramen and groove for the hyomandibular trunk, a mixed nerve containing fibers from cranial nerve VII plus the anteroventral lateral line nerve; iocn, infraorbital sensory canal; iop, interopercle; l, lacrimal, anteriormost infraorbital = io1; le, lateral ethmoid; mpt, metapterygoid; mptp, metapterygoid process; mx, maxilla; mxp, anterior articular process of maxilla; n, nasal; nop, posterior nostril; of, olfactory foramen; op, opercle; pa, parietal; pcl, postcleithrum; pmx, premaxilla; pmxnp, nasal process of premaxilla; po, postinfraorbitals, used here to refer to the infraorbital bone posterior to the orbit = io4 and io5; pop, preopercle; popcn, preopercular sensory canal; pro, prootic; pt, posttemporal; q, quadrate; qc, condyle of quadrate; ro, rostral bone; roc, opening to rostral sensory canal; sag, supraangular; scl, supracleithrum; sclcn, sensory canal of supracleithrum, which connects to the main trunk lateral line; smx, supramaxilla; so, subinfraorbital, used here to refer to the small infraorbitals between the lacrimal and the postinfraorbital = io2 and io3; socn, supraorbital sensory canal; sop, subopercle; spo, sphenotic; stcn, supratemporal sensory canal; su, supraorbital bones.

## 3. Results

### 3.1. Systematic Paleontology

Order AMIIFORMES Hay, 1929 [24]
Superfamily AMIOIDEA Bonaparte, 1839 [25]
Family AMIIDAE Bonaparte, 1839 [25]
Subfamily SINAMIINAE Patterson, 1973 [26]
*Khoratamia* gen. nov.
ZooBank LSID: urn:lsid:zoobank.org:act:B567BBC1-A881-4FC3-9ADA-DBD9828385D0
Type species. *Khoratamia phattharajani* gen. et sp. nov.; see below.
Diagnosis. As for the type and only species, *Khoratamia phattharajani* (below).

Etymology. *Khoratamia* is derived from "Khorat", the local name of the city and province of Nakhon Ratchasima, northeastern Thailand, which also gives the name to the Khorat plateau, + *Amia* (Greek).

*Khoratamia phattharajani* gen. et sp. nov.

ZooBank LSID: urn:lsid:zoobank.org:act:363AC015-83DC-4780-89CB-4D0A63438EF7

Holotype. NRRU-F01020023 (Figures 3–6). Anterior half fish with the dermal bones preserved.

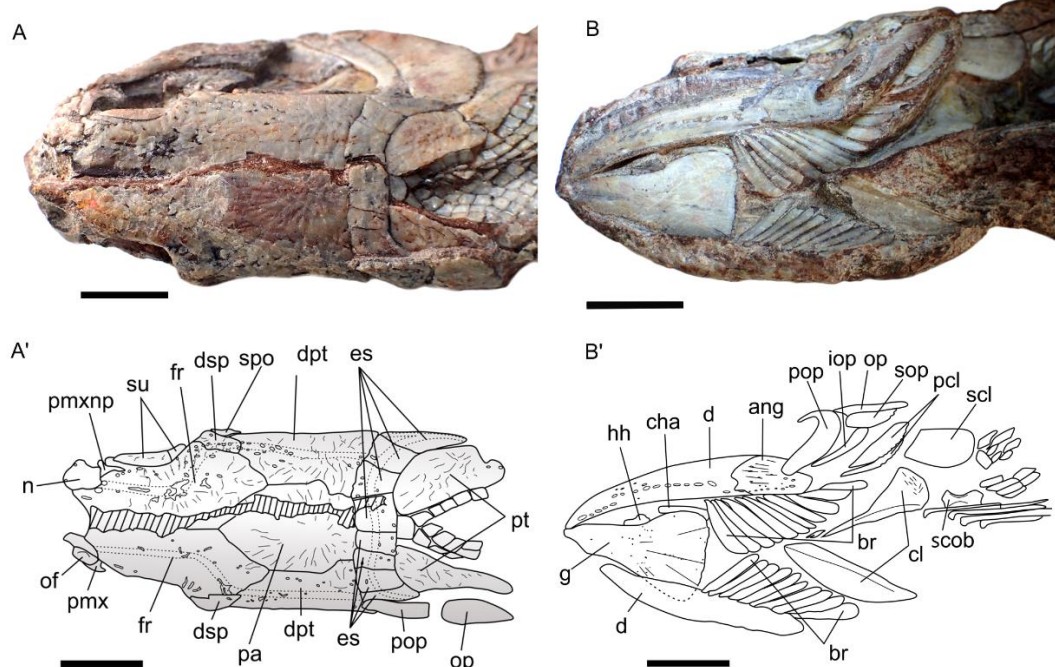

**Figure 3.** *Khoratamia phattharajani* gen. et sp. nov., holotype NRRU-F01020023, skull roof in dorsal view: (**A**) interpretative line drawing (**A'**) and ventral view (**B**) interpretative line drawing (**B'**). Scales bars: 1 cm.

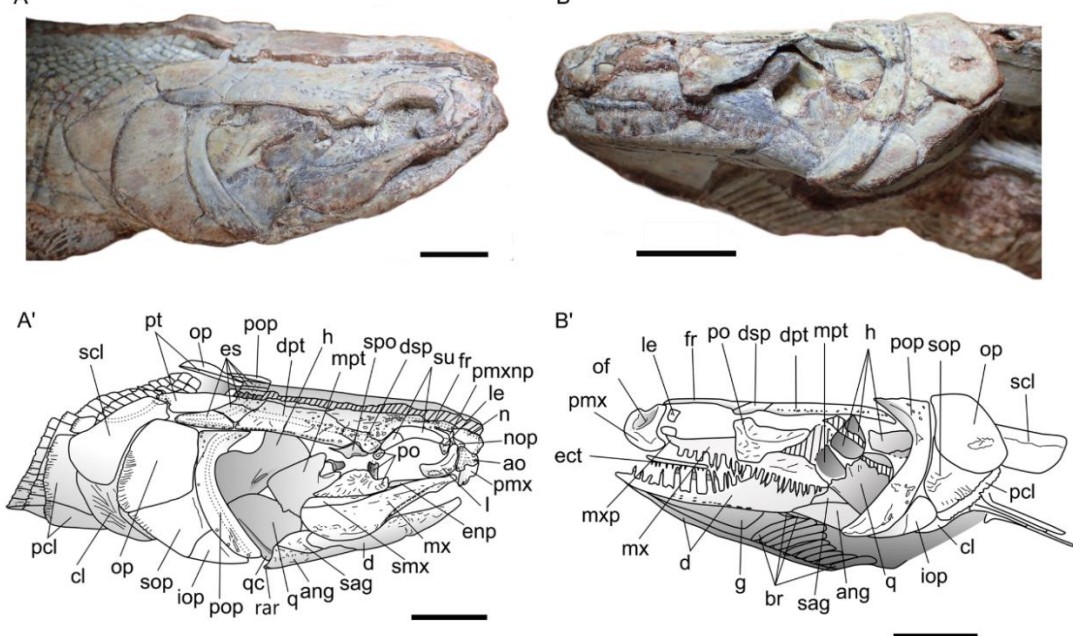

**Figure 4.** *Khoratamia phattharajani* gen. et sp. nov., holotype NRRU-F01020023, photograph of the skull in right lateral view: (**A**) interpretative drawing (**A'**) and left lateral view (**B**) line drawing (**B'**). Scales bar: 1 cm.

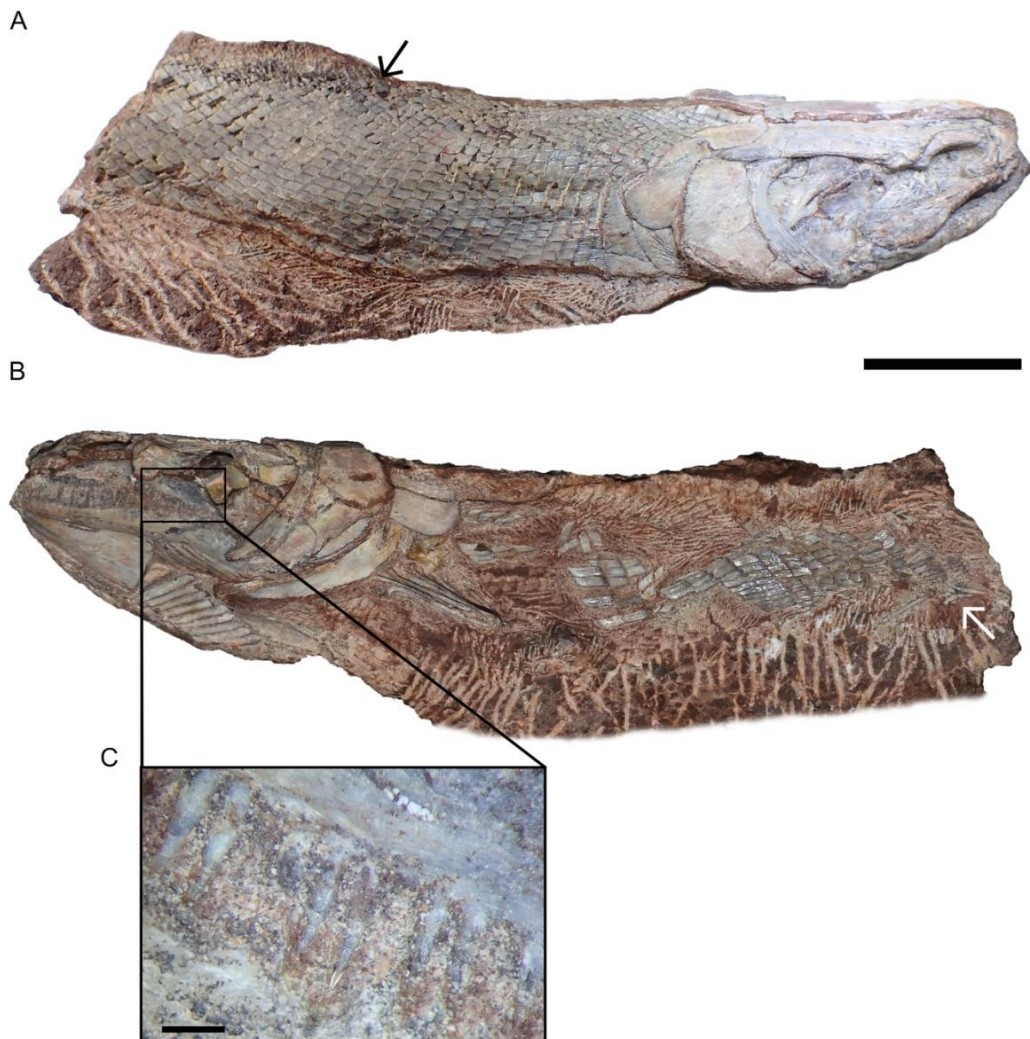

**Figure 5.** Holotype of *Khoratamia phattharajani* gen. et sp. nov., NRRU-F01020023: (**A**) photograph of the complete specimen in right lateral view, an arrow indicating the origin of the dorsal fin; (**B**) left lateral view, a white arrow indicating the position of the pelvic fin. The black box indicates the area enlarged in (**C**) (close-up of the teeth from a maxilla). Scales bars: 2 cm (**A**,**B**); 0.1 cm (**C**).

Paratype. NRRU-F01020024 (Figure 7), portion of a body with articulated scales, with a series of vertebral centra wedged in between both flanks of the specimen.

Diagnosis. Sinamiin fish characterized by the following combination of characters: crescent preopercular with a broad dorsal end; four pairs of extrascapulars with the six medial ones rectangular in shape and the lateralmost ones elongate and triangular in shape; unpaired parietal tapering anteriorly and narrowing at its posterior extremity; pyriform-shaped gular plate with a smooth undulate posterior margin; ventral postinfraorbital elongated and tapering posteriorly; small dorsal postinfraorbital; high cylindrical teeth with conical enamel stalk marked by fine ridges and with an arrowhead-shaped acrodin cap with cutting carinae; rhombic scales with serrated posterior margins; no peg and socket articulation of the scales with a keel on its internal surface at the center of the scales.

Type locality. Ban Krok Duean Ha, Tambon Suranaree, Amphoe Muang Nakhon Ratchasima, Nakhon Ratchasima Province, NE Thailand.

Type horizon. Khok Kruat Formation, Early Cretaceous, Aptian.

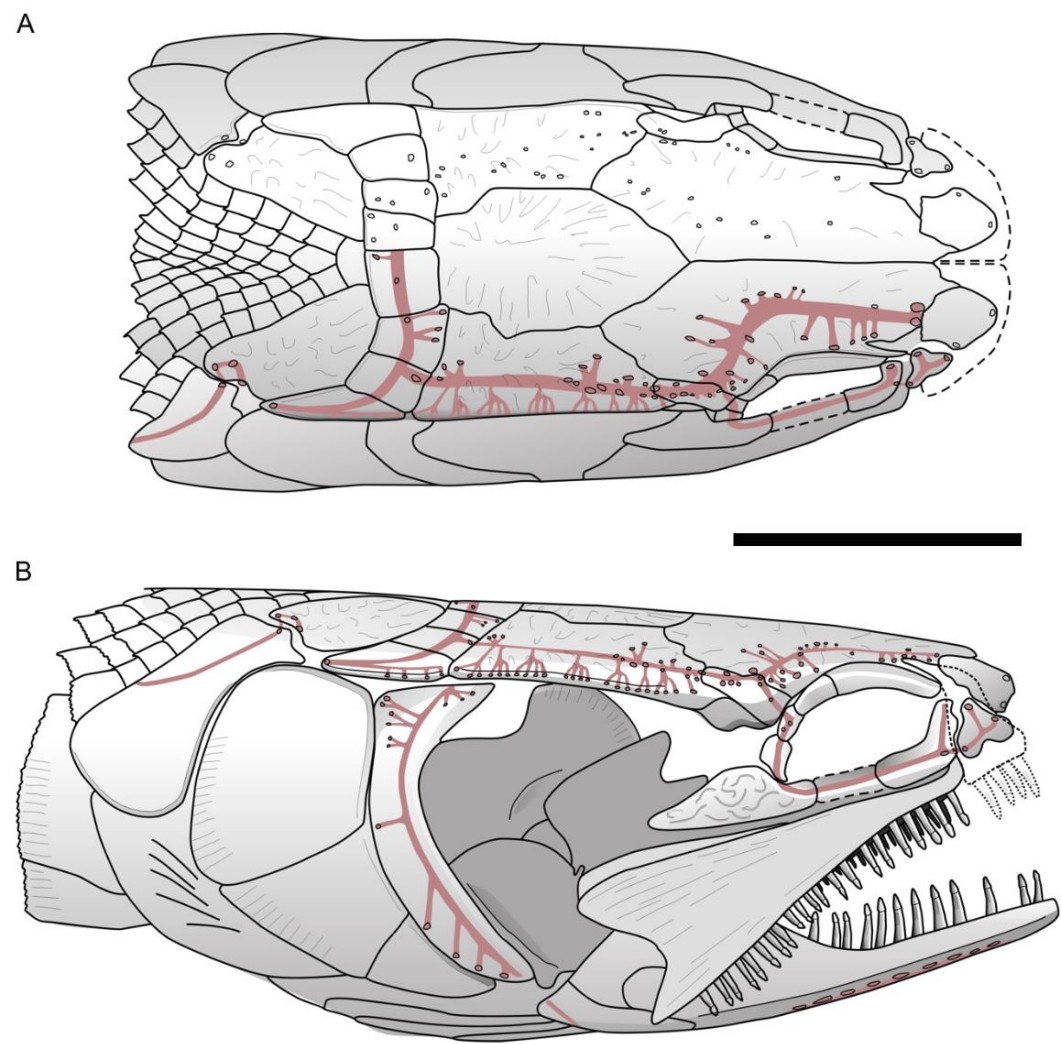

**Figure 6.** Skull reconstructions of a *Khoratamia phattharajani* gen. et sp. nov. showing the pattern of sensory canals in dorsal view (**A**) and lateral view (**B**). Scale bar: 2 cm.

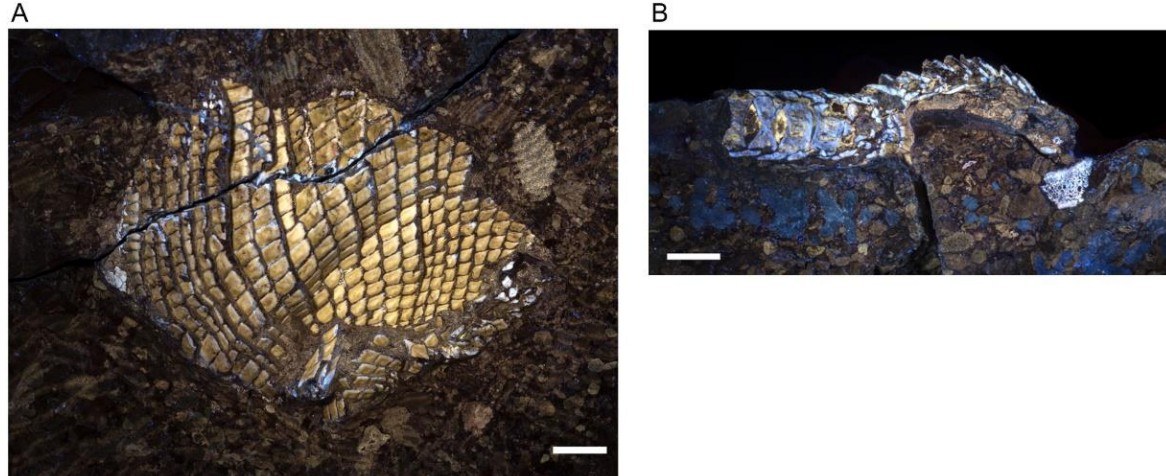

**Figure 7.** Paratype of *Khoratamia phattharajani* gen. et sp. nov., NRRU-F01020024: (**A**) specimen photographed under UV light; (**B**) series of vertebra column imaged with UV light. Scale bar: 1 cm.

Etymology. *Khoratamia phattharajani* derives from *Phatthara + Rajan* the designation of Somdet Phra Phatthara Maharat, a title given to King Bhumibol Adulyadej, (Rama IX) in recognition of his research dedication and support for breeding Nile tilapia (*Oreochromis niloticus*), which has provided a new career for over a million Thai agriculturalists and became a primary protein source for the Thai people. The discovery of a new sinamiin fish resulting from cooperation between Thailand and Japan commemorates the close collaboration between the two countries in fish research and aquaculture, with Emperor Akihito presenting 50 Nile tilapias to King Rama IX in 1965.

### 3.2. Description

### 3.2.1. General Features

The material at hand is not complete and only the head and part of the trunk can be described. The general feature of the fish is an elongated and cylindrical body shape based on the holotype. The head length is 45 mm (including the opercular series), and approximately 25 mm in depth. The ossifications of the head have no ganoin cover. Bones in the skull roof bear a faint rugose ornamentation. The body is covered by rhombic scales with their posterior margin serrated. The base of the anterior part of the dorsal fin is visible on the right side of the holotype, as well as elements of the paired fins on the left side of the same specimen.

### 3.2.2. Cranium

*Skull roof*. The frontals are the longest ossifications of the dermal skull roof. Each frontal is slightly narrower anteriorly than posteriorly, and marks a regular curvature at the level of the orbit. The posterior margin of the frontal is W-shaped and is firmly sutured with the paired dermopterotics and a median parietal. The anterior margin of the right frontal is notched. This notch is filled with a bone identified as the tip of the nasal process of the premaxilla. The lateral branch of the notch extends ventrally as a small process, with a concave surface that probably accommodated the dorsal extremity of the lacrimal (slightly shifted on the holotype). We cannot observe the suture between the left and right frontals on most of their length because of the presence of a crack along the mid-way of the skull roof. Only a short posterior portion of the suture presents a straight line towards the point extremity of the median parietal. The dorsal surface of the frontal bears a faint rugose ornamentation forming a reticulated pattern from the center of ossification. The frontal is twice as long as broad at its broadest posterior level. The parietal is a flat large median ossification as in other sinamiids. It is longer than wide (12 vs. 8 mm), with a sharp and pointed extremity wedged in the reversed V of the W-shaped margin of both frontals anteriorly. The right lateral margin is slightly undulate whereas the left lateral margin is straight. Both lateral margins mark a narrowing at the level of the posterior fourth of their length, thus forming a spearhead-shaped parietal. The dorsal surface of the parietal bears ridges radiating from the center of the bone. The dermopterotic borders laterally the parietal and extends anteriorly as a lateral pointed process with a slightly undulate suture along the posterolateral margin of the frontal. The dermopterotic is nearly three times longer than broad (14 vs. 5 mm), and as long, but narrower, as the parietal. Unlike the frontal and parietal, the dermopterotic is slightly convex in lateral view. Ridges radiating from the center of bone are present. The dermosphenotic is a small roughly rectangular bone wedged laterally between the right anterior extremity of the dermopterotic and the posterolateral margin of the frontal. It forms part of the skull roof. Its surface bears ornamentation similar to the rest of the skull roof bones. Posterior to the parietal and dermopterotics is a series of four pairs of extrascapulars. The six medial extrascapulars are rectangular in shape and the two most lateral ones are triangular with a blunt spine extending posteriorly. The external surface of the extrascapulars has no ornamentation, but small pores for the extrascapular commissure. The anteriormost preserved bone of the skull roof is the nasal, visible on the right side. The nasal is a plate-like ossification lying on the ascending nasal process of the premaxilla. It articulates with frontal posteriorly.

*Braincase*. Nothing is visible of the braincase, except the lateral extremity of the sphenotic wedged between the dermosphenotic and the dermopterotic, and fragments of ethmoid bones. The lateral ethmoid is visible on the right side as a vertical piece of bone held vertically under the anterolateral corner of the frontal, which forms a small ventral process (see above). On the left side, this region is not well preserved, and only a fragment of the lateral ethmoid is visible posterior to the nasal process of the premaxilla, which is almost vertical.

*Circumorbital Ring*. The circumorbital ring is well preserved on the right side. It consists of a dermosphenotic, two supraorbitals and three postinfraorbitals with the largest one located posteroventrally to the orbit, a lacrimal and an antorbital. Ventral to the orbit is a gap between the ventral most postinfraorbital and the lacrimal, in which lay probably one or two subinfraorbitals. The dermosphenotic ossification has no contact with the orbit, because anteriorly there is a tiny element corresponding likely to the third postinfraorbital. The shape of dermosphenotic is roughly rectangular with a notch at the anterodorsal corner of the bone. The external surface of the bone is not smooth and bears several pores located mostly along the dorsal edge. Two supraorbitals are present above the orbit. They are firmly sutured to each other and form a narrow crescent shape with their convex dorsal margins fitting well the deep curvature of the frontal leaving no space to the frontal to line the orbital space. Anterior to the orbit is the lacrimal, which is slightly shifted unveiling the lateral ethmoid underneath. It is boomerang-shaped, with the length of dorsal arm equal to ventral arm. Its concave dorsal margin forms the anteroventral border of the orbit. The antorbital, without contact with the orbit, is roughly triangular in shape with pore openings for the sensory canal at each summit of the triangle. A space delimitated by the antorbital, lacrimal, frontal and nasal corresponds to the posterior nostril. The lacrimal is separated by a small gap from the massive ventral postinfraorbital indicating that one or two small subinfraorbital were probably present on the living fish. The ventral postinfraorbital, situated at the posteroventral corner of the orbit, is the largest and more massive bone of the circumorbital ring. It bears strong ornamentation with strong knobs and grooves, even stronger than on the dermal roof bones. The shape is very peculiar, somewhat conch shell shaped, with the anterior portion protruding as a blunt process, with a broad mid-length area and with the posterior extremity tapering. The margins of the bone are smooth except the ventral one, which is undulate in correspondence to the ridges and grooves lying on the lateral surface. There are two small postinfraorbitals dorsally to the massive postinfraorbital. The second postinfraorbital is deeply sunk from its normal position. It is trapezoidal in shape with the ventral edge forming an oblique line contacting a curvature of the massive postinfraorbital. Its dorsal edge is indented at the mid-length for the path of the sensory canal. The third postinfraorbital is the smallest. It is rectangular in shape, and its surface bears a pore opening.

*Jaws*. The premaxilla is incomplete, being located in the most destroyed region of the skull. Therefore, its outline cannot be precisely described but the nasal process is apparently triangular in shape and held almost vertically. The extremity of the nasal process extends posteriorly as a spiny process fitting a notch on the frontal. The exposure of the nasal process on the skull roof is, as far as we know, unique within halecomorphs but present in lepisosteoidei among holosteans. On the right premaxilla, one broken tooth is visible on the oral margin of the bone, but the total number of teeth cannot be estimated. A large excavation marking the path of the olfactory nerve, with the nonvisible olfactory foramen located in the bottom is present on the left premaxilla. The maxillae are visible on both sides of the head. On the left side, the posterodorsal part of the bone is visible as an imprint only, but it clearly shows the concave margin that forms a deep notch. On the right side the posterior extremity of the maxilla appears distinctive convex, but the posteroventral part of the ossification is not preserved.

The oral margin of the maxilla is better preserved on the left side. It is slightly concave. There are about thirty-three teeth in total; each has a long conical base. The acrodine cap is very peculiar, being proportionally small with an arrow-head shape with cutting carinae.

There are some pits on the lateral surface of the right maxilla. A supramaxilla is preserved on the right side just above the posterior part of the maxilla. It is narrow and elongate, ca. 8.5 mm in length and 2 mm in depth at the posterior part, and gradually tapering towards its anterior extremity. The length of the bone is about half length of the maxilla.

The mandibles are low and their ventral margins expand horizontally. The visible part of the lower jaw comprises a dentary that bears a single row of conical teeth, an angular, a supraangular and a retroarticular visible as a tiny ossification at the posteroventral tip of the right mandible. The angular forms most of the posterior part of the mandible. It contacts the dentary along a zigzag suture anteriorly and the supraangular dorsally, which is partly visible on both sides. The dentary is long and anteriorly very shallow in lateral view with the posterior part of the bone increasing in depth to form the coronoid process. The dentary bears about 20 conical teeth with pointed arrowhead carinate acrodine caps. The teeth vary in size: the largest ones are located at the mid-length while the ones on the posterior and anterior portions of the bone are smaller. Each tooth is composed of a high cylindrical base and extends with conical enamel stalk marked by fine ridges. Its acrodine cap forms a rounded base and gradually tapers at the tip. Along the lateral edges of acrodine tip are strongly sharp carinae. All teeth are curved inwardly into the mouth.

*Suspensorium.* The well-preserved suspensorium on the right side of the holotype shows the hyomandibula, the metapterygoid, the quadrate and a piece of the entopterygoid visible in the gap present in the circumorbital ring, whereas fragments of these bones plus a piece of ectopterygoid with small teeth is present on the left side. The right hyomandibula exposes its lateral side. The main axis of the bone is only slightly inclined posteriorly from the horizontal axis, and only part of the rounded articular head, the concave posterior margin and a fragment of the concave anterior margin are visible. The foramen and groove for the hyomandibular trunk, a mixed nerve containing fibers from cranial nerve VII plus the anteroventral lateral line nerve, opens near the anterior concave margin of the bone. The general shape of the hyomandibula is discussed in the "comparisons" section. Anterior to the hyomandibula is the metapterygoid, which is roughly subrectangular, with a posterior portion slightly expanding over the hyomandibula and an anterior portion narrowing. Its anterior margin has a deep notch for the path of the trigeminal nerve, between the basal and otic processes, according to Stensiö's (1935) nomenclature [5]. Ventrally, it connects the quadrate through an interdigitate suture. The otic process of the metapterygoid is twisted while the basal process is blunt. The lateral surface of the ossification is totally concave. On the left side, the ectopterygoid is visible as a poorly preserved rod of bone with tiny recurved teeth. The quadrate is a large fan-shaped ossification. Its lateral surface shows a concavity extending the concavity on the metapterygoid. The convex ventral end forms the condyle for articulation with the lower jaw. Along the posteroventral border of the bone runs a strong crest starting from the condyle. The symplectic is not visible.

*Opercular series, branchiostegal rays, and gular.* The opercular series is complete, formed by the preopercle, operculum, subopercle and interopercle.

The preopercle is a narrow crescent-shaped bone with its extremities distinctly widening both ventrally and dorsally. The anterior and posterior margins are regularly curved, while the dorsal margin is straight. The preopercle is lying along the anterior edges of the opercle, subopercle and interopercle. The dorsoposterior edge of the bone forms a plan that marks an angle with the rest of the ossification and completely fits with the opercle. This feature seems to be peculiar to *Khoratamia* among halecomorphs. The anteroventral margin of the bone rests against the strong crest of quadrate, which runs along the upper arm. The opercle is roughly trapezoid in shape and deeper than long (12 × 8 mm). The dorsal and anterior borders are rather straight, while the posterior and ventral borders are slightly curved. The nearly upper half of the posterior edge is smooth whereas the lower part has strong serrations corresponding to the fine grooves present on its venteroposterior surface. The shape of the subopercle, lying between the opercle and interopercle, is somewhat the miniature of opercle in overturn position. It reaches the half depth of the opercle. The posterior edge of the subopercle is entirely serrated and no anterior ascending process is

visible. The interopercle is visible as a triangular bone wedged between the subopercle and the posteroventral edge of the preopercle. Its anterior tip is pointed and it runs toward the ventral tip of the preopercle. The ventral margin of the bone is gently convex.

Eleven elongated branchiostegal rays are well preserved on the holotype, KT-181; the posterior most one is the shortest but the widest. The branchiostegal rays are rather closely packed to each other. Each ray is narrow proximally and much broader distally. The anterolateral surface of each ray bears a crest, which is pronounced at the articulated point and gradually decreases and becomes complete flat at the half length of the bone. Anterior to the branchiostegals is a median pyriform-shaped gular plate. The anterior portion is longer and narrower than the posterior portion. On the mid-line of the ventral surface is a ridge that nearly reaches the center of the bone. The ventral surface of the expended posterior portion shows faint ridges and grooves of regular size extending to the posterior margin, which forms a slightly undulate edge. In the central portion of the gular are randomly distributed small pits. The ceratohyal is partly exposed between the left dentary and the gular plate, but its shape is hard to describe precisely. A thin element anterior to the ceratohyal is regarded as the hypohyal.

*Cephalic sensory canals*. The numerous pores at the surface of the bones indicate the general pattern of the sensory canal system. In some parts of the head, the path of the canals can be seen through the very thin bone. The supraorbital sensory canal opens via numerous pores on the dermopterotic and frontal. The dermopterotic bears approximately 12 irregularly arranged pores on its dorsal surface. Moreover, tiny pores also open along the thin lateral margin of the dermopterotic, as previously observed in other sinamiids. The connection with the preopercular canal is not visible and it is possible that it occurred on the ventral surface of the bone. On the frontal, the pores are irregularly arranged and more concentrated above the posterior level of the orbit and in the anterior portion of the bone. The main supraorbital canal apparently forms an angle in the center of the frontal and gives off tubuli medially and laterally that reach the openings spread over the surface. The dermosphenotic, which is part of the skull roof, bears ca. six foramens irregularly arranged. The path of the canal in the nasal is unclear, but a large pore is present on the anterior margin of the bone. The supratemporal commissure is clearly visible by transparency along the mid-length of the extrascapulars. The lateral most extrascapular, which extends posteriorly, bears ca. six openings along its lateral border in alignment with the series on the dermopterotic. On the posttemporal, two openings for the sensory canal are present on the lateral margin of the posterior prominent knob of bone, and a third smaller foramen opens between the two on the dorsal surface. The pores on the lateral margin of the bone correspond to the entry and exit of the canal, which thus forms a short loop in the posttemporal. This pattern implies that a section of the canal is not bone-enclosed between the exit of the extrascapular and the entry in the posttemporal. The sensory canal runs posteroventrally from the knob on the posterior extremity of the posttemporal to the mid-depth of the bone posteriorly. The preopercular sensory canal runs along the center of the bone. It gives off posterior tubuli that open via a few pores in the depressed posterodorsal corner, at least one pore in the mid-depth of the bone and at least four pores along the ventral part of the preopercle. The infraorbital sensory canal crosses both small dorsal postinfraorbitals as indicates the presence of pores and notch. We cannot follow the path of the canal within the large ventral postinfraorbital, in particular the potential presence of tubuli extending posteriorly as in *Sinamia zdanskyi* [5] because of the presence of strong ornamentation. There are pores located at the center of the lacrimal indicating the path of the canal that reach the antorbital anteriorly. The three pores in the antorbital indicates the path of a canal that crosses the bone and reaches the rostral (not preserved), and an opening dorsally close to the posterior nostril. The mandibular sensory canal runs along the ventral horizontal lamina of bone formed by the angular and dentary. Small foramens along the medial margin of the angular, which are followed anteriorly by a larger pore at the limit between this bone and the dentary, and by another medium-size foramen aligned in the dentary. Along most of the length of the dentary, the ca. 10 large

oval pores are shifted laterally compared to the posterior alignment, and located at the limit between the ventral and lateral sides of the mandible. At the anterior extremity of the mandible, five large oval foramens are located again along the medial margin.

3.2.3. Postcranium

*Pectoral girdle and fin.* The large posttemporal lies posteriorly to the extrascapulars. It is visible as a large trapezoidal to ovoid shaped ossification with a notch along its posterior margin for the path of the sensory canal. Its external surface bears a faint rugose ornamentation forming a reticulated pattern from the center of ossification. The anteroventral edge is strongly convex whereas its dorsal edge is almost straight with slight undulation. Because of the posterior extension of the lateral most extrascapulars, the posttemporals do not reach the lateral margin of the skull roof and have no direct contact with the opercle. The supracleithrum is proportionally very large, with an elongated general rectangular shape, an expended ventral extremity and a dorsal border firmly applied against the notch at the posterior margin of the posttemporal. Along the dorsal margin of the supracleithrum runs a short and curved ridge, which lies against a notch at the posterior margin of the posttemporal. The cleithrum is well visible on the right side. It is crescent in shape with the anterior part narrower than the posterior part. The lateral face of the cleithrum is marked by smooth ridges that converge to the center of the expanding posterior area. The ventral margin of the cleithrum is convex. Two large postcleithra are preserved in connection on the right side and slightly shifted on the left side. The upper postcleithrum is larger and deeper and is prominently convex dorsally and becomes thinner in its ventral part, which is flattened. The lowest postcleithrum is small and less convex, but it is not complete, making the estimate of its shape impossible. The posterior margins of the postcleithra are serrated with grooves extending on the surface of the bone. A saddle-shaped ossified scapulocoracoid is partly visible on the left side.

Eleven fin rays are visible, but more were likely present. The preserved portion of the rays, ca. 25 mm long, are unsegmented, indicating that the total length of the fin was proportionally longer because about half of the rays are unsegmented in the pectoral fin of *Amia calva*.

*Vertebral column.* A series of vertebrae is visible on the paratype (NRRU-F01020024). Five articulated centra are preserved wedged between the left and right scales covering of the body. The length of each centrum is shorter than its depth in lateral view (ca. 6 mm long and 11 mm deep, respectively). Owing to two broken posterior centra in their middle part, the sandglass shape of the centrum is visible and indicates that the anterior and posterior articular surfaces are deeply concave. The lateral surface present trace of grooves and ridges.

*Dorsal fin.* Although this area was destroyed, the origin of the dorsal fin can, however, be located. It is placed at the 5/8 distance of the pelvic fin.

*Pelvic girdle and fins.* The pelvic fins are apparently very small based on the fin rays partly preserved on the left side, which are small and short. The basipterygium is visible as a long rod with an enlarged posterior extremity.

*Squamation.* The whole body is covered by thick rhombic scales covered with a ganoin layer. The scales of the dorsal region are small, as long as deep, whereas the scales of the lateral and ventral regions of the trunk are distinctly longer than deep (about 3 mm long and 1.5 mm deep). In addition, their lateral surface has a ridge along the ventral edge in the holotype, but in KAB-30 (NRRU-F01020025) there are two ridges located near the dorsal and ventral margins. The posterior margin of all scales is serrated, with about 11 tiny denticulations in maximum (the number varies from 2 to 11 depending on the scale position on the body). The lateral line scales are not visible on the anterior part of the preserved body. At the level of the beginning of the dorsal fin, a scale every two or three scales is marked by a sensory pore, which has the shape of a vertical elongate furrow dug in the middle of the scale. No peg and socket articulation has been observed in the area where scales are altered. The imprints of scales indicate that the internal surface of the scales

protrudes like a strong keel or ridge at its center. The keel is visible on scales preserved in internal view behind the dorsal fin of the holotype.

### 3.3. Comparison

*Khoratamia phattharajani* shares a mosaic of anatomical characters with its closest relatives, the sinamiins, but also with other amiid subfamilies that merit discussion before a phylogenetic analysis is conducted. In amiins, the nasal process of the premaxilla slopes backward, in line with the dorsal contour of the skull roof in lateral view. In addition, the nasal process is wide in its posterior part. In contrast in *K. phattharajani*, as in *Siamamia* (Figure 8D), the nasal process of the premaxilla is maintained almost vertically, making a marked angle with the contour of the skull roof, and the nasal process tapers posteriorly. This arrangement is more similar to that of the vidalamiin, *Calamopleurus* (compare Figure 8A,B,F).

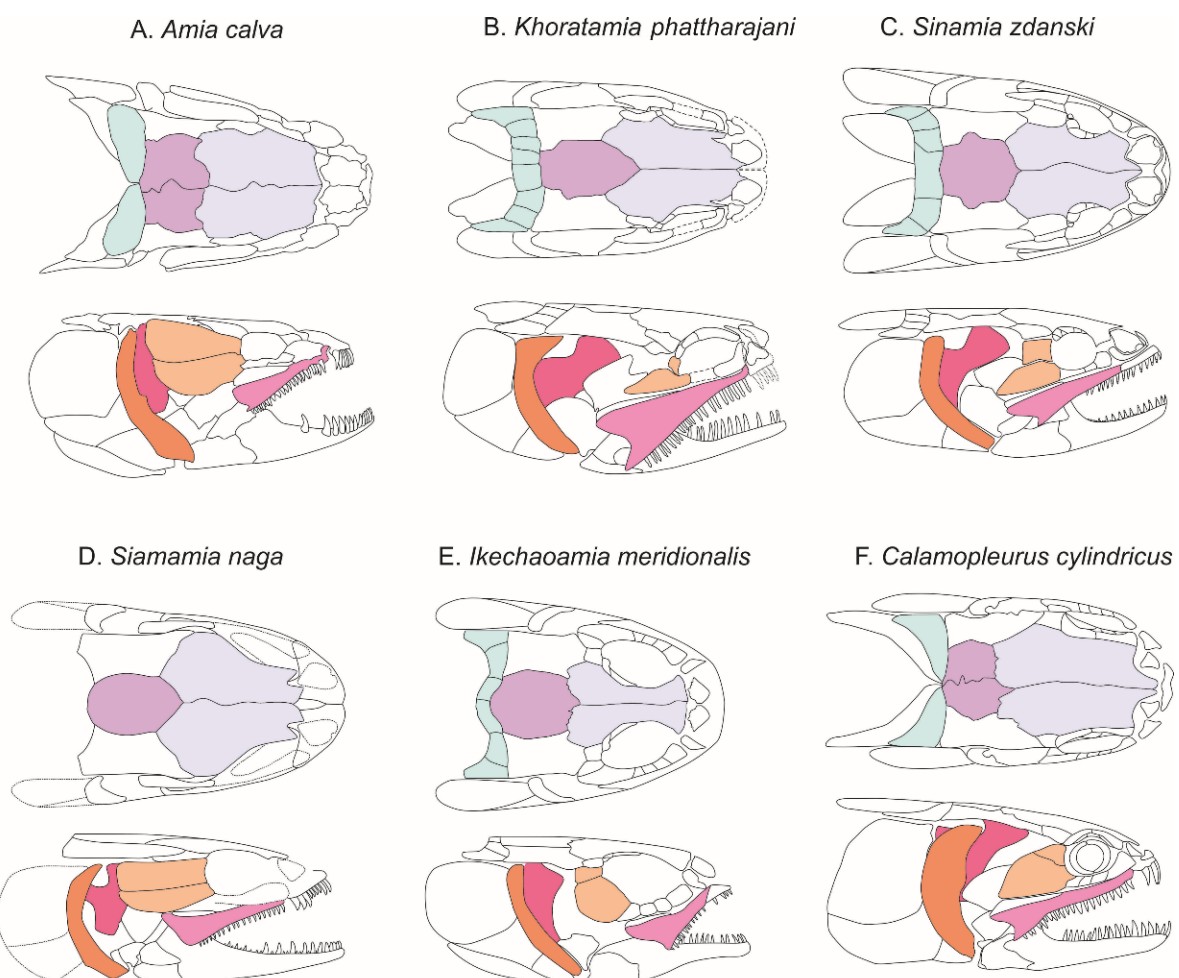

**Figure 8.** Comparison of reconstructed skulls of various amiids in dorsal and lateral views. The colored bones are those discussed in the text.

Although poorly preserved, the posterior end of the maxilla of *K. phattharajani* is deep and deeply notch, a shape closer to the maxilla shape of the vidalamiins (compare Figure 8A,B,F), although the ventral process below the notch is not as elongated. The keeled acrodine tip of the teeth of *K. phattharajani* is also a feature shared with all vidalamiins but absent in amiins and other sinamiins.

The preopercle of *K. phattharajani* has a narrow crescent shape forming a more pronounced anterior curvature than in most other amiids and differs in its distinctive broadened dorsal end, the surface of which marks an angle with the rest of the body of the bone

(Figure 8). The preopercle of *Sinamia lanzhouensis* [11] has a similar general shape, but the dorsal end are narrower than those of *K. phattharajani*.

In *K. phattharajani*, the median parietal (Figure 8) has a spearhead shape with a pointed anterior end and inwardly curved lateral margins at the posterior fourth, while in *S. zdanskyi* the anterior part of the bone is blunt and the lateral margins show broad obtuse processes inserting into the concavities of the medial margins of the dermopterotics. Such a process, although smaller, is visible on the right lateral margin of *K. phattharajani*. The lateral edges of the parietal are regularly concave in *Siamamia naga* [16], and in *Ikechaoamia* [3].

The large ventral postorbital located at the posteroventral angle of the orbit of *K. phattharajani* is similar to the fifth infraorbital of *S. liaoningensis* [10] and the fourth infraorbital of *S. zdanskyi*. However, the bone in *K. phattharajani* has a somewhat peculiar conch shell shape (Figure 8B) unlike the trapezoidal shape of *S. zdanskyi* (Figure 8C) and the rectangular shape of *S. liaoningensis*.

A pyriform-shaped gular has been described for two isolated ossifications of *Sinamia kukurihime* [15], whose outline resembles that of *K. phattharajani*, except for the smooth posterior margin of the first. The Japanese specimens, however, are not completely preserved and the comparison between the two taxa is still uncertain. Although having a similar piriform shape, the outline of the gular plate of *S. zdanskyi* differs from that of *K. phattharajani* by being proportionally much broader toward the posterior end. *Calamopleurus cylindricus* has a gular plate with a concave lateral margin, somewhat similar to the gular plate of *K. phattharajani*, but in the former the concavities are shallower and the posterior margin of the bone is strongly denticulate.

The scales of *K. phattharajani* are characterized by the absence of a peg and socket articulation, the presence of denticulations along the posterior margin, and a strong keel that marks the inner side, all characteristics which differ from *Siamamia naga*. This feature helps distinguish the two genera when they are only represented by isolated scales in some fossil sites.

The sensory canal system of *K. phattharajani* (Figure 6) is characterized by a higher pore density than in *Amia*. A typical feature is the presence of a series of aligned pores along the lateral edge of the dermopterotic and the most lateral extrascapular. The pores of the extrascapular are apparently connected to a canal that starts from the supratemporal canal near its exit from the extrascapular. It is not known how the lateral openings of the dermopterotic are connected to the infraorbital canal. On the left side, a broken part of the lateral margin of the frontal shows a group of three openings leading to a single small canal reaching the suborbital canal. In *Sinamia zdanskyi*, a series of pores along these two bones is also present, and Stensiö [5] (Text-fig 18) suggested that the pores along the lateral edge of the dermopterotic are connected to the main canal via groups of two or three tubuli. In *Siamamia*, a similar series of pores opens along the lateral margin of the dermopterotic [16]. In both Thai species, the supratemporal sensory canal also opens on the surface of the dermopterotic, whereas such openings are apparently absent in *A. calva* and *S. zdanskyi*. In *Sinamia liaoningensis*, there is a high density of sensory canal openings along the lateral margin of the dermopterotic, and especially along the lateral extrascapular, but few on the dorsal side of the dermopterotic [10]. This feature, when visible, appears to be restricted to sinamiids among halecomorphs.

*3.4. Phylogenetic Analysis*

The result of the parsimony analysis is as follow: three most parsimonious trees (Figure 9), with a length of 258 steps; consistency index (CI) equals 0.4961; retention index (RI) equals 0.7368; and rescaled consistency index (RC) equals 0.3656. The list of characters selection from Xu [4] and are explained in detail in Supplementary Materials.

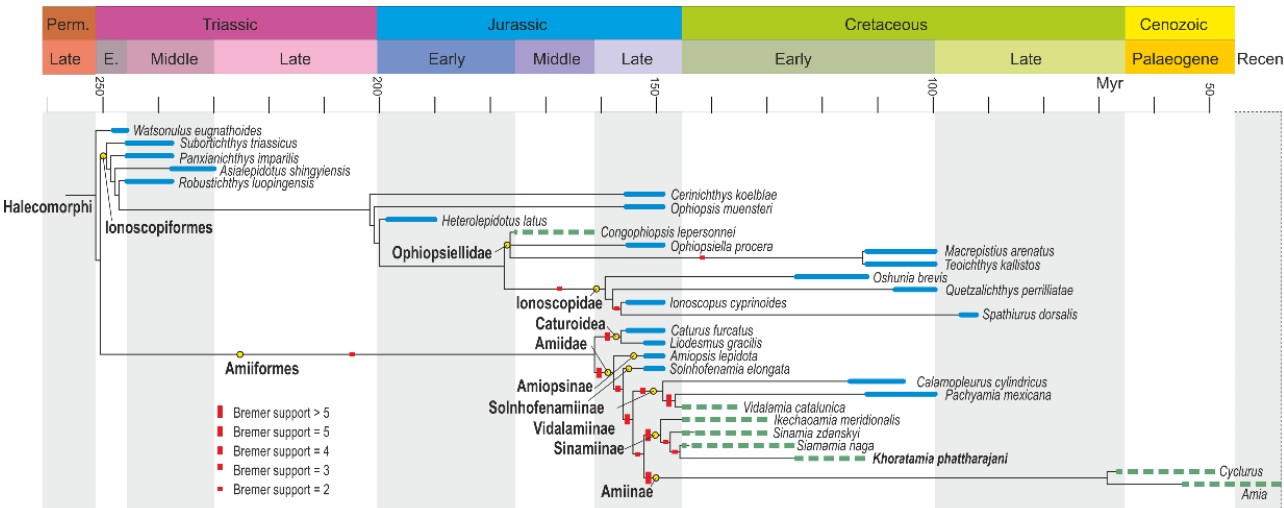

**Figure 9.** Strict consensus tree of the three most parsimonious trees: 258 steps; consistency index (CI) equals 0.4961; retention index (RI) equals 0.7368; and rescaled consistency index (RC) equals 0.3656. Stratigraphic ranges of marine taxa are shown in blue; green dotted lines indicate the stratigraphic ranges of continental taxa.

The resulting strict consensus tree is similar to that obtained in Xu [4], except for the nodes modified by the addition of the two Amiiformes, *Khoratamia phattharajani* and *Siamamia naga*, and similar to that obtained in El Hossny et al. [23] for the same taxa used as the latter study did not include *Cyclurus kehreri* among the amiiforms. Owing to the similarities in the resulting trees, we only discuss the nodes relevant to the two newly added Sinamiinae. Nodes within Amiiformes are more strongly supported than nodes within Ionoscopiformes, with Bremer support ranging from two to more than five, depending on the node.

The two Thai species are recovered as sister taxa, and both recovered as sister group to *Sinamia zdanskyi*, and *Ikechaoamia meridionalis* as sister to them, all of them forming the clade Sinamiinae. Thirteen characters support this node (4, 6, 9, 14, 15, 53, 77, 81, 82, 84, 98, 114, 117), with a single autapomorphy, the "presence of an unpaired parietal" (9). A second character state is present only at this node, "six or less principal caudal fin rays below lateral line in adults" (114, with $ci = 0.667$, but this is because there are two derived states), but it should be noted that the state is unknown in the two Thai taxa. The character state "more than two pairs of extrascapular bones" (15) is present in all sinamiins where this feature is known, but also in two basal Ionoscopiforms from the Triassic of China (*Subortichthys triassicus* and *Asialepidotus shingyiensis*). The "presence of an elongation of the opercular process of the hyomandibula" (53) and "the absence of a well-developed anterodorsal process of the subopercle" (77) are character states shared with a few other halecomorphs, while the other supporting character states are shared in parallel with more taxa. Of note, the "presence of rhomboic scales" (117) supporting the node of sinamiins is the only observed reversal from the "amioid-like scales" state that appears at the base of Amiiforms. Transformations from the rhomboic scale to the amioid scale also characterize the Ionoscopids.

Sinamiins and amiins are grouped on the basis of a single autapomorphy, "absence of sclerotic ring" (44) and by two homoplasies (19 and 26).

Within the sinamiins, the node uniting *Khoratamia* and *Siamamia* is supported by twelve characters (10, 18, 19, 20, 22, 23, 25, 28, 50, 79, 87, 118), all homoplastic and most of them ambiguous.

## 4. Discussion

*Systematic Affinities and Implications on the Evolutionary History of the Lineage*

Stensiö [5] included *Sinamia* among the Amiidae, then Berg [27] coined the Sinamiidae, a generally accepted family in later studies. Grande and Bemis [3] produced a phylogeny that resolved the Amiidae, containing the Amiinae, Vidalamiinae, Solnhofenamiinae and Amiopsinae, together forming the sister family of the Sinamiidae. Patterson [26], however, coined the subfamily Sinamiinae, which he considered the sister to the Amiinae within the Amiidae. In this study, however, not all the taxa considered by Grande and Bemis [3] were considered. Recently, Xu [4] tackled the neopterygian interrelationships and, based on a new set of characters and a new data matrix, he resolved the Sinamiinae as sister to the Amiinae, with the two other subfamilies (Vidalamiinae, Solnhofenamiinae) as sister to this pair. This is the phylogenetic pattern that we obtained here, probably because our data matrix is largely based on the same data set as Xu [4]. Accordingly, we refer the four genera of this clade (*Sinamia*, *Ikechaoamia*, *Siamamia* and *Khoratamia*) to the subfamily Sinamiinae.

While support for the Sinamiinae clade is strong, its grouping with the Amiinae is still weakly supported. In particular, the return from amioid-like scales to rhomboic scales may seem surprising from an evolutionarily point of view. To our knowledge, no other similar examples are known in the evolutionary history of ray-finned fishes whereas the reverse—shift from rhomboid scales to amioid-like scales—has occurred several times in the evolutionary history of halecomorphs [4]. Both clades, amiins and sinamiins, are strictly freshwater as far as we know, unlike most basal Amiiformes that were marine dwellers. It cannot be excluded that parallel adaptation to freshwater environments favors the evolution of morphological convergent traits that alter the phylogenetic signal. However, Bremer's support for all amiiform in-group clades is strong, and we assume for the moment that amiins and sinamiins are in fact sister clades.

The temporal range of sinamiins is older (Early Cretaceous) than the temporal range of amiins (mainly Cenozoic), and while the geographical distribution is restricted to Eastern Asia for the former, the geographical distribution of the amiins cover the whole northern hemisphere [3]. The ghost lineage of amiins is very long, almost the entire Cretaceous interval, and we cannot exclude that the Cretaceous amiin taxa—still unknown—exhibited character combinations that root the lineage in the sinamiins.

If this hypothesis is true, the Asian origin of amiins would have been followed by dispersals throughout Eurasia and towards North America via Berhingia; similar dispersal patterns were observed in several lineages of ray-finned fish during the Cretaceous–Paleogene time interval, such as in the Acipenseriforms, some Osteoglossomorphs and the Cypriniformes [28].

## 5. Conclusions

The discovery of *Siamamia* in the Sao Khua Formation at the Phu Phok site extended the geographical distribution of sinamiins much farther south than their previous distribution, mainly in China. Then, the discovery of isolated sinamiin remains in several localities in Sao Khua Formation, and in the younger Khok Kruat Formation, indicated that these fish formed an important component of the continental aquatic vertebrate assemblages during the Early Cretaceous in what is now Thailand. The discovery of articulated material of a new genus and a new species of sinamiin in the Khok Kruat Formation, in a sister position to *Siamamia*, is an indication that diversification of this lineage had occurred in Southeast Asia, regardless of diversification in China. Note that another sinamiin fish specimen from the Phu Phok site, figured in Cavin et al. [17], may correspond to a third distinctive taxon, thus reinforcing this region as a potential location for diversification of the clade.

The phylogenetic relationships obtained here, indicating the existence of a possible lineage of freshwater amiids formed by amiins and sinamiins, suggest that the extant freshwater *Amia* lineage may have originated in Asia before dispersal to North America.

**Supplementary Materials:** The following supporting information can be downloaded at: https://www.mdpi.com/article/10.3390/d15040491/s1.

**Author Contributions:** Conceptualization, U.D. and L.C.; methodology, U.D., W.N., Y.N. and H.Y.; software, T.E.H. and L.C.; validation, U.D., W.N., P.J., Y.N., H.Y., T.E.H. and L.C.; formal analysis, L.C. and T.E.H.; investigation, U.D., W.N., P.J., T.E.H. and L.C.; resources, U.D., W.N., P.J., Y.N., H.Y. and L.C.; data curation, U.D., W.N., P.J., Y.N., H.Y., T.E.H. and L.C.; writing—original draft preparation, U.D. and L.C.; writing—review and editing, U.D., W.N., P.J., Y.N., H.Y., T.E.H. and L.C.; visualization, U.D. and L.C.; supervision, L.C.; project administration, U.D., W.N., P.J., Y.N., H.Y. and L.C.; funding acquisition, U.D., W.N., P.J., Y.N. and H.Y. All authors have read and agreed to the published version of the manuscript.

**Funding:** This paper is a contribution to the project "Palaeoecology of Thai Mesozoic freshwater bony fish" by the Office of the Higher Education Commission and the Office of Research Fund—Developing the Potential for Research Work of New Generation Teachers, fiscal year 2018 (MRG6180086).

**Institutional Review Board Statement:** Not applicable.

**Informed Consent Statement:** Not applicable.

**Data Availability Statement:** Not applicable.

**Acknowledgments:** This work was supported by Thailand–Japan Dinosaur Excavation Project, a collaborative research project between the Northeastern Research Institute of Petrified Wood and Mineral Resources, Thailand, and the Fukui Prefectural Dinosaur Museum, Japan. We would like to thank all the members of the project who took part in field work and prepared specimens. We express our thanks to Masateru Shibata, Teppei Sonoda, Duangsuda Chokchaloemwong, Nareerat Boonchai and Supanut.Bhuttarach for their support. We also thank Bouziane Khalloufi (Palaeontological Research and Education Centre) for his dedicated help in photographing specimens under UV light, as well as the two anonymous reviewers for their constructive comments. This work was also supported in part by the Office of the Higher Education Commission and the Office of Research Fund—Developing the Potential for Research Work of New Generation Teachers, fiscal year 2018 (MRG6180086) to U.D., and the International Research Group PalBioDivASE (IRN) grant of CNRS.

**Conflicts of Interest:** The authors declare no conflict of interest.

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
