# Peer review of "A New Sinamiin Fish (Actinopterygii) from the Early Cretaceous of Thailand: Implications on the Evolutionary History of the Amiid Lineage"

_diversity, doi:10.3390/d15040491_

Round 1

Reviewer 1 Report

The manuscript is an interesting and scientifically relevant contribution to the knowledge about the diversity and phylogenetic relationships among sinamiins (amiids), thus it is worth to be published. The description is complete, and the figures are of high quality as far as it can be judged from the pdf file. Data for the phylogenetic analysis are adequate and up to date and results are robust. The taxonomic placement of the new genus as a sinamiin is well supported and the discussion about the biogeographic evolution of the group is undoubtedly worth to be further investigated in subsequent studies. I strongly recommend acceptance.

My other comments (see annotations in the attached pdf) mostly concern some revisions on the English spelling and expression that would be easy to fix.

Author Response

Comments and Suggestions Responses
Line 621. amioid like scales have independently evolved at least twice in halecomorphs.

We modified the sentence as follows: “In particular, the return from amioid-like scales to rhomboic scales may seem surprising from an evolutionarily point of view. To our knowledge, no other similar examples are known in the evolutionary history of ray-finned fishes whereas the reverse - shift from rhomboid scales to amioid-like scales - has occurred several times in the evolutionary history of halecomorphs [4]”

Line 626. generating a wrong clade, this sentence is unclear for me

We modified two sentences as follows:

“It cannot be excluded that parallel adaptation to freshwater environments favors the evo-lution of morphological convergent traits that alter the phylogenetic signal. However, Bremer's support for all amiiform ingroup clades is strong, and we assume for the mo-ment that amiins and sinamiins are in fact sister clades.”

Reviewer 2 Report

Most of the figures are difficult, to read. Can they be enlarged and can the line drawings be sharper? They lines are not sharp enough. Fig. 6, 6c is a difficult one and fig. 8 is unreadible.

I there a picture of the paratype with scale(s)?

Please change the name, don't name a fossil after a former king, that's highly inappropriate in a scientific paper.

Author Response

Here is a point-by-point response to the reviewer’ comments and concern.

Comments and Suggestions

Responses

Line 112. what kind of element? scales?.

Done

Line 140- anatomical terminologies

Done

Line 179- add the picture of NRRU-F01020024

We added figure 7 in the text

Line 184. ?? is something missing here?

Done

Line 195. specific etmology, please a neutral name not a king’s name

As far as we know, no article of the International Code of Zoological Nomenclature prevents naming a species with the name of a king. Several examples already exist, with several names inspired from the Thai Royal family. In the revised version, however, the specific name was modified as: “Khoratamia phattharajani is derived from Phatthara + Rajan refer to Somdet Phra Phatthara Maharat, a designation of Somdet Phra Maha Bhumibol Adulyadej, the great King Rama IX, Thailand”

Line 261. ventral to the orbit is a gap between the ventral most postinfraorbital and the lacrimal, in which lay probably one or two subinfraorbitals

This pattern is presented in the reconstruction figure

Line 297. of or within?

Done (within)

Line 312. what do you mean, straight or concave?

Done (slightly concave)

Line 317. correct ‘long’ to be length

Done

Line 347. 1935, which paper, the same as the one cites already?

We added the reference in brackets [5]

Line 363. I don't understand this, what is this space?

Done

Line 460-vertebral column. I advise to include a figure of this specimen too

Done (figure 7)

Line 486- figure 6. can't see much in C. All other details are really small in this figure and difficult to judge

We modified the new version in the text and change to figure 5

Line 508. what do you mean with internal curvature

Change to; -a narrow crescent shape forming a more pronounced ‘anterior’ curvature

Line 511. which tips?

Done

Line 598- figure 8. this figure is unreadable

Done change to figure 9

Line 641. why surprising?

Delete and correct
